# Autochthonous Cultures to Improve Safety and Standardize Quality of Traditional Dry Fermented Meats

**DOI:** 10.3390/microorganisms11051306

**Published:** 2023-05-17

**Authors:** Franca Rossi, Patrizia Tucci, Ilaria Del Matto, Lucio Marino, Carmela Amadoro, Giampaolo Colavita

**Affiliations:** 1Istituto Zooprofilattico Sperimentale dell’Abruzzo e Molise (IZSAM), Sezione di Campobasso, 86100 Campobasso, Italy; 2Dipartimento di Medicina e Scienze della Salute “V. Tiberio”, Università degli Studi del Molise, 86100 Campobasso, Italy

**Keywords:** traditional dry fermented meats, protection policy, desired microbiota, lactic acid bacteria, coagulase negative staphylococci, autochthonous cultures, safety, sensory quality

## Abstract

Traditional dry fermented meat products are obtained artisanally in many countries, where they represent a gastronomic heritage well distinguished from industrial counterparts. This food category is most often obtained from red meat, a food commodity that is under attack because of evidence of increased risk of cancer and degenerative diseases with high consumption. However, traditional fermented meat products are intended for moderate consumption and gastronomic experience, and, as such, their production must be continued, which would also help safeguard the culture and economy of the geographical areas of origin. In this review, the main risks attributed to these products are considered, and how these risks are reduced by the application of autochthonous microbial cultures is highlighted by reviewing studies reporting the effects of autochthonous lactic acid bacteria (LAB), coagulase negative staphylococci (CNS), *Debaryomyces hansenii* and *Penicillium nalgiovense* on microbiological and chemical safety and on sensory attributes. The role of dry fermented sausages as a source of microorganisms that can be beneficial to the host is also considered. From the results of the studies reviewed here it appears that the development of autochthonous cultures for these foods can ensure safety and stabilize sensory characteristics and has the capacity to be extended to a larger variety of traditional products.

## 1. Introduction

Fermented meat products include dry sausages, minced fermented meats with alternative surface coverings and sliced salted meats fermented without stuffing. These products comprise hundreds of traditional variants whose artisanal production has been transferred between generations as a practice to preserve meat during the year for family sustenance, as happens today in rural areas across the world. More often, traditional fermented meat products are manufactured on an artisanal scale in small manufacturing plants in their geographical areas of origin.

Traditional fermented meat products represented a precious source of many essential nutrients, micro-nutrients, vitamins and minerals for past generations [1,2]. These are mostly obtained from red meats, i.e., mammalian muscles [3] and, as such, should be consumed with moderation to prevent an increasing risk of various cancers and chronic diseases attributed, though not with full agreement, to high amounts of red meat consumption [4,5,6,7,8]. Indeed, the health problems attributed to eating red meat are probably determined by the constant increase in per capita amounts consumed worldwide since the 1960s [9].

Raw fermented and dry meats in European countries are represented mainly by dry sausages that, as other traditional food products, are protected by policies that aim to avoid the loss of cultural heritage, gastronomic experience and economic income in marginal areas. According to the European legislation, the term ‘traditional’ designates products with proven usage on the domestic market for at least 30 years, a period that allows transmission between generations [10]. However, most traditional fermented meat production processes date back several centuries [11], as reported in the production norms publicly available, for example, from the Italian ministry of agriculture [12].

The protection of traditional products in European countries is currently accomplished according to the EU Regulation 1151/2012 [10], which sets the rules for the introduction of traditional foods in the European register of products with protected designation of origin (PDO) and protected geographical indication (PGI), publicly accessible at https://ec.europa.eu/info/food-farming-fisheries/food-safety-and-quality/certification/quality-labels/geographical-indications-register/ (accessed on 13 March 2023). Entering a product name in the register is promoted by producer associations and approved by the European Union (EU) [13]. From this register it is possible to observe that Portugal, with 23 products, is the EU country with the highest number of PGI/PDO dry fermented sausage types, followed by Italy with 17 products. Countries with a few raw sausages in the register are Spain, France, Germany, Hungary, Poland, Romania, Croatia, Slovenia and Cyprus, while for Finland a few products from cold smoked raw reindeer meat are included.

Countries also protect traditional products that have not achieved the PDO or PGI status. For instance, each year the Italian ministry of agriculture updates a public list of national traditional products among which are numerous dry fermented sausages typical of each Italian region [14]. Italian cured meat products are protected further by a national decree that imposes the requirement to specify on the label the country/countries in which the animals were born, raised and slaughtered [15].

Since the production costs of traditional dry fermented meats are high, producers continue to manufacture traditional foods in strict compliance with operating quality schemes, dictating extensive farming and particular races and feeds for the meat-producing animals, as long as commercial competition is fair [10,16]. Therefore, PDO and PGI and other protection labels, such as the Slow Food Presidium [17], are law instruments that demonstrate to consumers the authenticity and quality of traditional products.

The manufacturing processes of fermented sausages are technologically simple, not requiring cooking steps or extensive changes in the structure and composition of the raw materials. Raw meat is only minced or chopped, mixed with fat in different ratios and flavored with salt, spices and/or other natural ingredients (e.g., garlic, wine, potatoes, aromatic herbs, juniper berries, orange peel). Finally, the mixture is stuffed in natural (i.e., parts of animal intestines) or synthetic casings and let dry in appropriate conditions, often local seasonal temperature and relative humidity (RH) parameters, for a period that depends on the diameter of the product but is of at least one month. The Italian Pitina sausage is not stuffed in casings but covered uniformly with maize flour [12]. Some fermented meat products that can be consumed raw are not stuffed in casings, e.g., products manufactured in African, South American and Asian countries, including El-Guedid, or «el khlî», an Algerian meat product made from sheep, beef, goat or camel meat cut into strips, seasoned with salt and spices, exposed to the sun until reaching a_w_ 0.625 and pH 5.2–5.5 and finally stored in jars for up to one year [18], the Algerian Kaddid [19], beef jerky [20], Biltong in South Africa [21], Kilishi, a jerky-type RTE sun-dried meat product from Northern Cameroon and Nigeria [22,23], Kitoza from Madagascar [24] and Socol and Charqui from South America [25,26].

The simple manufacturing processes do not justify comparing these products to extensively processed meats undergoing cooking phases and/or other intense technological treatments. However, as for most PDO and PGI sausages the addition of nitrate and nitrite salts and sugars, or a sugar source such as milk powder, is recommended to ensure safety by nitrite formation and acidification by lactic acid bacteria (LAB) [12,27]. The presence of nitrites represents a controversial safety issue for these products [28].

The sensory characteristics of traditional products must comply with the specifications described in the official norms of production. These carry out audits to establish the degree of products’ conformity to guarantee product homogeneity during the year and in different years, as well as the distinction from similar products on the basis of multivariate chemometric approaches and sensory parameters [10,29]. The production norms dictate the towns of production, animal races and farming practices, permitted feedstuffs, meat cuts to be used, product manufacturing phases, permitted additives, appearance and flavor of the end product and number of LAB and coagulase negative staphylococci (CNS) at the end of ripening as a hygiene indicator. Examples of recommended values are at least 100,000 CFU/g LAB for Salame di Felino and 10^7^ CFU/g mesophilic microbiota, comprising LAB and CNS, for Salame Brianza [12]. The usage of microbial starter cultures is prescribed for many protected sausage types. A detailed definition of the starter cultures is given for Salame Napoli, which has been proposed for the PGI status, for which the number and type of selected cultures that it is possible to add—namely lactobacilli, *Micrococcaceae*, staphylococci and pediococci—are specified [30]. It is also specified that cultures able to reduce nitrates and not too much acidifying must be used. Indeed, the final pH of sausages from Southern Europe ranges between 5.1 and 5.79 [27,31,32]. However, for some products the addition of starter cultures is not prescribed, and safety is deemed to be accomplished exclusively by modulating environmental conditions and exploiting the natural microbiota [12]. The inhibition of most pathogens and spoiling microorganisms in these products is determined by the decrease in the a_w_ to as low as 0.843, as in the examples of the Italian Ventricina del Vastese sausage, at the end of ripening [33], the decrease in pH, and the presence of NaCl and nitrites.

The origin, composition and selection process of the microbial cultures mentioned in the production norms are not specified. Therefore, producers can choose between commercial starter cultures or selected autochthonous cultures. For instance, the use of a commercial starter culture including *Latilactobacillus sakei*, *Pediococcus acidilactici*, *Staphylococcus carnosus* and *S. carnosus* subsp. *utilis* was tested in a Sardinian traditional fermented sausage and found to control *L. monocytogenes* growth *Enterobacteriaceae* without affecting the numbers of LAB and CNS or the product’s composition. However, the effect of that mixed culture on the product’s sensory characteristics was not evaluated [34].

On the other hand, the presence on the market of producers of selected starter cultures offering customer-tailored services for the development of autochthonous cultures indicates that producers, or producer associations, are interested in using cultures which can preserve the distinctive organoleptic characteristics of their products.

In this review, studies regarding the use of autochthonous microbial cultures to standardize the safety and quality of dried fermented meats are summarized with the scope of showing the potential of components of the natural microbiota to reduce safety risks while maintaining sensory distinctness. The activities of the main microbial groups involved in the ripening of these products are described, and how autochthonous representative strains of these groups counteract particular safety concerns and exalt the sensory parameters of traditional fermented dry meats is presented.

## 2. Technologically Relevant Microorganisms in Dry Fermented Meats

The safety and sensory and nutritional quality of dry fermented meats rely upon the prompt development at appropriate levels of microbial groups comprising different species of LAB and CNS.

### 2.1. LAB Occurring in Dry Fermented Meats

LAB contribute to the safety of dry fermented meats by producing mainly lactic acid from carbohydrates and lowering pH, thus increasing the inhibitory activity of salt and drying towards pathogenic and spoiling microorganisms. *L. sakei* is the species that predominates in fermented meat products, followed by *Lactiplantibacillus plantarum* and *L. curvatus* [35]. *L. sakei* is particularly well adapted to the meat ecological niche for its efficient utilization of the substrates available. One of these is arginine, an abundant amino acid in meat that *L. sakei* metabolizes through the arginine deiminase pathway (ADI), encoded by the *arc*ABCTDR—*PTP* gene cluster, with the production of ornithine, ammonia and carbon dioxide. Concomitantly, ATP is generated, thus providing a source of energy for this species. Moreover, *L. sakei* is able to ferment sugars present in meat such as ribose and the pentose moiety of nucleosides, as well as *N*-acetyl-neuraminic acid. *L. sakei* is equipped with genes that favor its survival in oxidative stress conditions, namely a heme-dependent catalase *cat* gene and a manganese/iron superoxide dismutase gene *sod*A. Tolerance to high salt concentrations and acidic pH is strain-dependent. Genetic heterogenity has been described for this species, with three intra-species clades distinguished by multilocus sequence typing (MLST) [36]. In a study regarding the traditional product Ventricina del Vastese, 70 *L. sakei* isolates were grouped in different Rep-PCR genotypes of which only one lowered the pH to 5.3 in meat extract suspension with no subsequent increase, thus indicating that only some naturally occurring *L. sakei* strains are able to improve product safety by matrix acidification [37].

*L. sakei* strains and other LAB that occur in fermented meats produce bacteriocins able to inhibit *L. monocytogenes* and *S. aureus*. In particular, *P. acidilactici* strains produce pediocins that inhibit the growth of pathogens such as *L. monocytogenes* and *C. perfringens* [36,38,39,40], and a number of studies regarded the effects of LAB cultures producing anti-listerial bacteriocins in dry fermented sausages with promising results [41].

Some adventitious LAB form excess hydrogen peroxide that may cause discoloration and lipid oxidation. CNS starter cultures can limit the oxidative process with their catalase (CAT) activity [38].

### 2.2. Coagulase Negative Staphylococci in Fermented Dry Meats

Coagulase-negative staphylococci (CNS) are commonly found on animal skin and mucosae from which they contaminate meats and meat manufacturing plants [42]. CNS’s main roles are promoting safety by reducing nitrate salts (NO_3_^−^), used as additives in these products, to nitrites (NO_2_^−^) able to inhibit *Clostridium botulinum.* With this reaction CNS also promote the development of a desired red color given by the combination of myoglobin and nitric oxide (NO) deriving from the spontaneous reduction of nitrites to form nitrosomyoglobin (MbFe^II^NO) [43]. Inoculation with the CNS species *S. xylosus* confers a brighter red color on dry fermented sausages than the autochthonous microbiota. Moreover, lower lipid oxidation is observed, possibly due to the antioxidant capacity of nitrites [44]. In general, all species belonging to the CNS group possess nitrate reductase activity, with the exception of *S. succinus* subsp. *succinus*, one of the species associated with European fermented sausages [11,30,45]. However, NO can also be produced from arginine, which is abundant in meat, by the nitric oxide synthases (NOS) encoded by *nos* genes present in all CNS genomes, that can contribute to the formation of nitrosomyoglobin [43].

CNS are not in the list of microorganisms with qualified presumption of safety (QPS) status from the European Food Safety Authority (EFSA) [46] because hazardous genetic traits can be found in some strains. Risk characteristics to be examined in these bacteria are staphylococcal enterotoxin genes, panton-valentine leukocidin, toxic shock syndrome toxin-1, exfoliative toxin genes that can be horizontally transferred from coagulase positive staphylococci, biofilm and biogenic amine (BA) formation and antibiotic resistance (AR) genes. Other traits to be excluded are hemolytic and DNAse activity [47,48]. PCR tests can be useful in screening a high number of strains to exclude the presence of unwanted genes [47], but whole genome sequencing (WGS) is required to evaluate the most promising candidates for use as starter cultures for absence of hazardous traits, beyond AR testing, which is required for QPS species [49].

An example of a selection scheme for CNS starter cultures was reported by Sun et al. [48], who carried out a selection of CNS strains based on sarcoplasmic and myofibrillar proteins hydrolysis, salt and nitrite tolerance, nitrate reductase and lipolytic activity, absence of BA production and AR among 143 isolates. Then, the selected isolates, identified as *S. simulans* and *S. saprophyticus*, were characterized by WGS to demonstrate the absence of hazardous genetic traits.

CNS, like LAB, influence the flavor and aroma of dry fermented meats by forming compounds derived from their proteolytic and lipolytic activities that are strain-specific since enzymatic endowment varies at the strain level. In fermented sausages, protein hydrolysis, beyond originating precursors to the aroma compounds, facilitates water release, which favors the drying process [50].

### 2.3. Yeasts and Molds in Fermented Dry Sausages

The yeast species *Debaryomyces hansenii*, often associated with fermented sausages, comprises strains with the capacity to inhibit lipid oxidation and increase the production of volatile acid compounds [50,51] and aroma compounds derived from amino acid degradation [52].

Finally, in dry fermented meats with surface mold development, *P. nalgiovense* is the predominant species and has never been reported to produce mycotoxins [53]. This mold species confers flavors from proteolysis and lipolysis and replaces naturally occurring mycotoxigenic fungi [54]. For some products a smoking procedure is carried out during drying to prevent the growth of molds on the surface [55].

### 2.4. Diversity of Naturally Occurring Technologically Relevant Microorganisms in Dry Fermented Meats

Studies on the microbial ecology of traditional fermented sausages not inoculated with starter cultures have highlighted a high amount of variability in the natural microbiota among and within products, and this makes it difficult to achieve microbiological safety and uniform quality. The natural evolution during ripening of the technologically relevant microbial groups, LAB, CNS and yeasts, differs among products but also from batch to batch, as illustrated by the examples shown in Figure 1 for Catalão and Salsichão Portuguese sausages [27] and Ciauscolo Italian sausage [56].

These microbial groups have also been found to predominate in non-stuffed products such as Kitoza, in which LAB and CNS showed average counts of 6–7 Log CFU/g [24]. A high level of diversity in LAB and CNS species was reported for Mediterranean fermented sausages analyzed by phenotypic and molecular methods including new generation sequencing (NGS). The dominance of *L. sakei* and *S. xylosus* was observed in most products. Among LAB, the commonly occurring species of lactobacilli were *L. sakei*, *L. curvatus* and *Lactiplantibacillus plantarum*, but other species of lactobacilli, such as *Enterococcus*, *Leuconostoc*, *Pediococcus*, *Lactococcus*, *Tetragenococcus* and *Weissella*, were also identified [18,32,56,57,58]. *P. pentosaceus* was dominant in Salame Piemonte, as shown by metataxonomic analysis [32]. The CNS species most often identified in traditional fermented sausages were *S. xylosus*, *S. succinus*, *S. equorum* and *S. saprophyticus*. However, other CNS species, i.e., *S. cohni*, *S. carnosus*, *S. epidermidis*, *S. pasteuri*, *S. hominis*, *S. capitis*, *S. vitulinus* and *S. warneri*, were detected in some products [18,41,57,59,60,61]. In Salame Piemonte, moreover, *S. carnosus*, a species most often found in sausages from Northern Europe with lower pH values [59,60], was identified [32].

Yeast microbiota was mainly constituted by *D. hansenii*, but other fungal species were identified less frequently [32].

## 3. Safety Concerns in Traditional Fermented Dry Meats

The safety of dry fermented meats can be compromised by the microbiological and chemical hazards described below.

### 3.1. Microbial Pathogens

Microbiological hazards, based on the most recent alerts [https://ilfattoalimentare.it/argomenti/richiami-e-ritiri, accessed on 25 February 2023], are the major food pathogens *L. monocytogenes* and *Salmonella* serovars. According to the latest report on zoonoses in EU member states, fermented sausages were one of the foods with the highest occurrence of *L. monocytogenes* at distribution (3.1%). This pathogen caused an increasing number of infection outbreaks, with 2183 confirmed cases of invasive listeriosis in humans and the highest number of deaths in outbreaks notified to EFSA [62].

The results of a study on the survival of *L. monocytogenes* in Mediterranean-style dry fermented sausages highlighted that, despite its constant decrease during shelf life, the pathogen was still detected in the final product, showing the capacity to cope with hurdles occurring in the product [63]. Therefore, the competition by LAB, a_w_ and pH values must ensure unfavorable conditions for the growth of this pathogen [64]. The use of starter cultures proved to be a valid solution to the *L. monocytogenes* threat in raw fermented meats, as demonstrated by recent investigations. For example, in Čajna sausage, counts of four intentionally added *L. monocytogenes* strains belonging to serotypes 4b and 1/2a decreased significantly in all analyzed samples and were below the detection limit on day 18. Moreover, from day 7 until day 14, the decrease in *L. monocytogenes* was significantly faster in sausages in which commercial starter cultures comprising *D. hansenii*, *L. sakei*, *P. acidilactici*, *P. pentosaceus*, *S. carnosus* and *S. xylosus* were added [65].

One of the species most active against *L. monocytogenes* is *L. sakei*, in which different bacteriocins with anti-listerial activity were described, namely sakacin A (curvacin A), sakacin P (bavaricin); sakacin 674, sakacin K, sakacin V18, sakacin M (lactocin S), bavaricin MN, sakacin T, sakacin G, sakacin X, sakacin Q, sakacin 1 and sakacin G2, most of which are class II bacteriocins [66]. Therefore, a better exploitation of sausage-adapted LAB species by the focused selection of strains inhibiting *L. monocytogenes* should be further pursued.

### 3.2. Nitrosoamines

A controversial safety concern attributed to dry fermented sausages regards the presence of nitrates and nitrites. These are not carcinogenic by themselves but give rise to reactive nitrogen species, including nitric oxide (NO), that combine with secondary amines to form carcinogenic nitrosoamines [67]. Despite the fact that the main sources of dietary nitrates and nitrites are of plant origin, it was hypothesized that nitrates/nitrites in meat determine an increase in cancer risk for the presence of amines, amides and heme iron that favor the increased production of N-nitroso carcinogens. Consequently, there is a trend to reduce or eliminate the use of nitrates and nitrites in meat products [28].

Levels of added nitrites below 150 mg/kg are considered safe, and this is the maximum amount admitted in raw cured meats by the European law [68]. However, the levels of nitrites detected at end ripening in fermented sausages are much lower than this value and were reported to be less than 10 ppm, since their concentration decreases after their reaction with myoglobin [69]. The most recent re-evaluation of the safety of the nitrite levels added to foods by the European Food Safety Authority was based on the estimation of the formation of nitrosoamines inside the body following nitrite consumption. It was concluded that consumer exposure to nitrites and nitrates when used as food additives is within safe levels for all age groups. However, if all dietary sources of nitrites and nitrates are considered, the acceptable daily intake (ADI) may be exceeded for all age groups [70]. It must be underlined that the nitrosation reaction is inhibited by substances such as ascorbic acid [71], which is among the permitted additives in some Italian PDO and PGI fermented sausages [12].

There are a few available studies on the amount of nitrosoamines present in fermented sausages. One of these found low levels of N-nitrosodimethylamine (NDMA), N-nitrosodiethylamine (NDEA), N-nitrosodi-n-propylamine (NDPA), N-nitrosopyrrolidine (NPYR), N-nitrosopiperidine (NPIP) and N-nitrosodi-n-butylamine (NDBA) nitrosoamines in the Turkish Sukuc fermented sausage, in other sausage types, in salami and in cooked meat (Doner kebab). NDMA and NDEA are commonly found in sausages but at levels below the limit of 10 μg/kg set by some food safety agencies [71]. In a study carried out on kimchi *L. sakei* and *L. curvatus* were found to efficiently lower amounts of NDMA and were found to directly degrade NDMA during culture in MRS broth containing NDMA [72].

Nitrite unintentionally added to meat products derives from environmental contamination and can also contribute to the formation of nitrosoamines. One study evaluated whether a vegetable source of nitrates, namely chilli pepper used as an ingredient in Ventricina del Vastese sausage, led to excessive levels of nitrites in the sausage, and it was found that the maximum allowed amount of nitrates and nitrites was never exceeded in the product [73].

### 3.3. Biogenic Amines

Hazardous compounds that can be formed in fermented meats are BAs deriving from the decarboxylation of free amino acids by different bacterial groups. One of these groups is the genus *Enterococcus*, which produces both tyramine and β-phenylethylamine by the tyrosine decarboxylase (TDC) [74]. Enterobacteria are the main producers of cadaverine and putrescine [75]. In addition, BAs can be formed from strains of LAB belonging to different species for the presence of amino acid decarboxylase gene clusters that can be horizontally transferred (HGT) [76]. Therefore, the absence of decarboxylase genes must be ascertained by molecular tests in bacterial strains to be used as starter cultures [41].

Tyramine and histamine are the most toxic BAs, and the European Food Safety Authority (EFSA) has indicated a daily intake for healthy individuals of below 50 mg for histamine and below 600 mg for tyramine [77]. BA-degrading bacteria may also occur in fermented sausages and should be considered for use as starter cultures [78].

### 3.4. Mycotoxins

Another safety concern that may occur in fermented meats is the presence of mycotoxins, mainly ochratoxin A. Indeed, in some products molds are let develop on the surface to confer flavors from proteolysis and lipolysis [54]; the natural occurrence of molds able to produce mycotoxins in vitro was observed. This hazard is prevented by inoculating selected molds on the sausage surface to outcompete adventitious mycotoxin producers [53,79]. Delgado et al. [80] showed that a strain of *P. chrysogenum* inhibited the production of the mycotoxin cyclopiazonic acid, possibly present in dry fermented sausages beyond OTA, when co-inoculated with the mycotoxin producer *P. griseofulvum* on the sausage surface.

### 3.5. Polycyclic Aromatic Hydrocarbons

The smoking process carried out for some dry fermented sausages inhibits the growth of molds on the surface and also confers sensory connotations. It is usually carried out by direct contact between the product and smoke deriving from wood combustion. The released volatile compounds (VOCs) diffuse into the product through the permeable casings and confer aromas and tastes. However, among the VOCs developed during smoking, the polycyclic aromatic hydrocarbons (PAH) are recognized carcinogens, so that a safe level of 12 µg/kg for these compounds, expressed as the sum of the four substances, benzo(a)pyrene, benz(a)anthracene, benzo(b)fluoranthene and chrysene (PAH4), was fixed for smoked meat [81]. The amounts of PAHs found in smoked meat depend on the type of wood used for smoking [55]. Strains of LAB able to subtract PAHs have been applied to lower the amount of these compounds in fermented sausages, with good results in sensory evaluation as well [82].

## 4. Effect of Autochthonous Microbial Cultures on Safety and Quality of Dry Fermented Sausages

According to the protection rules, for traditional products recipes and manufacturing processes must be followed faithfully, and the possible interventions aimed to ensure safety must be respectful of product sensory characteristics and peculiarities. The role of the LAB and CNS microbiota on overall hygiene is crucial, but safety promotion and technologically relevant characteristics, such as pH lowering capacity, proteolytic activity and pathogen inhibiting capacity, are highly variable among native bacteria [35,37]. Therefore, components of the autochthonous microbiota, adapted to the ecological niches of traditional fermented sausages, must be selected by taking into account both safety assurance and the preservation of sensory distinctness.

Studies available on the application of LAB and CNS cultures isolated from traditional sausages as starters has often, though not always, led to safety and organoleptic quality improvements. Those retrieved from the databases of international scientific literature Google Scholar (https://scholar.google.com/, accessed on 20 February 2023) and Scopus (https://www.scopus.com/search/form.uri?display=basic#basic, accessed on 18 February 2023) are reviewed below. The main findings of the studies are summarized in Table 1 for the effects observed on safety and in Table 2 for the effects observed on sensory characteristics.

### 4.1. Effects of Autochthonous Cultures on Safety of Dry Fermented Sausages

In many of the studies consulted, both safety aspects and sensory evaluation were considered. As safety aspects, the inhibition of pathogenic microorganisms, mainly *L. monocytogenes*, or proteobacteria, which are both pathogenic and indicators of poor hygiene, were examined. The effect of starters on BA formation was also analyzed.

In the study by Li et al. [44], CNS with high protease activity were isolated from the traditional naturally fermented sausage Qianwufu. Strain *S. simulans* QB7, showing the highest proteolytic activity, tolerance to salt and nitrite concentrations and an absence of virulence genes and hemolytic, decarboxylase, DNAse and biofilm-forming activities, was used as a starter. This strain reduced the growth of undesirable bacteria, determined higher contents of total free fatty acids and free amino acids and lowered the pH and a_w_ values because of enhanced LAB development.

Baka et al. [83] evaluated the effect of inoculating five autochthonous LAB strains on the evolution of microbial groups, lipid oxidation, BA content and sensory attributes in a Greek fermented sausage. Treatment with *L. sakei* 4413 lowered the content of all Bas, with the reduction of tyramine by 13%, tryptamine by 55%, cadaverine by 60% and putrescine by 72%.

*L. curvatus* 54M16 isolated from a traditional fermented sausage of the Campania region, Italy, found to carry the genes for the bacteriocins sakacin X, T and P, was able to inhibit the pathogens *L. monocytogenes* and *Bacillus cereus* and the meat spoilage bacterium *Brochotrix thermosphacta.* The strain showed a good acidifying capacity and was able to hydrolyze sarcoplasmic proteins and lipids and reduce nitrates. Moreover, it showed high values of SOD activity and the formation of l-arginine, l-valine, l-phenylalanine and l-lysine free amino acids. In sausage fermentation it dominated the LAB population at all sampling times and lowered Enterobacteriaceae counts by about 1 Log at the end of ripening [84]. When the anti-listerial activity of this strain was analyzed in sausage production, it was found to be low for pathogen levels of about 4 Log CFU/g. However, *L. monocytogenes* naturally present in the raw ingredients was totally inhibited. As demonstrated by 16S rRNA-based metagenome analysis, this strain dominated and affected the bacterial ecosystem, inhibiting spoilage bacterial genera *Brochothrix*, *Psychrobacter*, *Pseudomonas* and some Enterobacteriaceae. The inoculated sausages presented a more intense ripened flavor in sensory analysis [88].

Kamiloglu et al. [91] also observed a 2 Log CFU/g reduction in *L. monocytogenes* in Sukuk sausage inoculated with an autochthonous *L. plantarum* strain compared to the not-inoculated control.

*S. xylosus* SX16 and *L. plantarum* CMRC6, isolated from Chinese Dong fermented pork Nanx Wudl, showed high proteolytic activity in a preliminary screening in vitro. The combination of the two strains (starter LS) suppressed the growth of Enterobacteriaceae and accelerated acidification and proteolysis during ripening, improving the content of total free amino acids and essential amino acids Phe, Ile and Leu.

In a study aiming to evaluate the effects of different associations of autochthonous starters, comprising the previously characterized *S. equorum* S2M7, *S. xylosus* CECT7057, *L. sakei* CV3C2, *L. sakei* CECT7056 and the yeast strain 2RB4 [75,85,86], on the safety and quality of Paio do Alentejo, a traditional Portuguese dry-fermented sausage, the starters were inoculated at high concentrations, and an extended fermentation step of 72 h was introduced before stuffing [87]. LAB and CNS in inoculated samples reached 3 Log CFU/g and 2 Log CFU/g higher counts, respectively, compared to the control. LAB remained at almost constant numbers throughout ripening, while CNS significantly decreased at the end of ripening, as commonly observed in dry fermented sausages [41,103]. Yeast and molds maintained similar numbers throughout the ripening time with no differences among treatments. *L. monocytogenes* was present in most analyzed samples, possibly due to persistence in the manufacturing plant or because it was originally present in the raw materials. However, in the final products its numbers were below the European legal limit of 100 CFU/g [64] for most treatments, and the starter cultures had significantly reduced *L. monocytogenes* counts. Moreover, the extended fermentation time enabled lower pH and a_w_ values to be reached, thus controlling the spoiling and pathogenic microbiota with the stabilization of enterobacteria. The latter had initial numbers of 5–6 Log CFU/g in the meat batter, indicating a poor hygienic quality in the raw materials [87].

The reduction of the total BA content in products inoculated with autochthonous cultures was also observed [87]. Vasoactive BAs tryptamine, β-phenylethylamine, histamine and tyramine showed mean values higher in control sausages (279.11 ± 131.94 mg/kg), but differences with inoculated products were not statistically significant. The use of starters had a significant effect on the content of total BAs, with lower values in inoculated sausages (between 452.58 ± 55.11 mg/kg and 533.96 ± 63.65 mg/kg) compared to control sausages (731.95 ± 206.23 mg/kg), with the lowest mean values in sausages inoculated with *S. xylosus* CECT7057 and *L. sakei* CECT7056. BA reduction compared to the not inoculated control by using autochthonous cultures had been obtained previously [85,86]. However, BA levels increased in all batches during ripening, except for β-phenylethylamine and tyramine, which was in not in line with previous findings [86]. Tyramine contents were lower than 2.20 mg/kg in all samples, while other authors observed much higher levels of this BA [104,105]. Different trends in BA formation can be explained by the predominance of bacterial groups producing different BAs among the native microbiota [87]. Regarding histamine, it was present at detectable but not hazardous levels in end products, similarly to what had been reported previously [86,104,106]. Tryptamine was the BA with highest concentrations in the products, as also observed in Catalão and Salsichão sausages [27] and in non-inoculated Serbian sausages [95]. This BA mainly contributed to the amount of total vasoactive amines that was higher than 200 mg/kg in end products. However, the total content in BAs was below 1000 mg/kg, showing an improvement compared to previous reports [86,107]. The presence of the natural polyamines followed the pattern usually observed in fermented sausages, with the prevalence of spermine.

Tyramine-degrading *S. epidermidis*, *L. sakei*, and *L. curvatus* were isolated from Harbin sausage [78]. When the *S. epidermidis*/*L. sakei* association was used in Harbin sausage production it reduced the tyramine content of the final product by 55% compared to the control, despite the high proteolytic activity of the *L. sakei* starter [108]. The effect of inoculation on the microbiota composition was examined by 16S rRNA gene metagenomic analysis, and it was found that lactobacilli were the most abundant bacterial group in Harbin dry sausage during fermentation in all samples. The inoculation of *L. sakei* or *S. epidermidis* separately increased the abundance of lactobacilli, while, when in association, the inoculated bacteria decreased the relative abundance of *Weissella* spp. that was found to be predominant in Harbin sausage [95].

Martín et al. [93] used *L. sakei* 205, harboring the pediocin PA gene [109] in a challenge test with *L. monocytogenes* in Salchichón traditional Spanish sausage. LAB counts in the *L. sakei* 205 inoculated samples were 7 Log CFU/g until day 30 and remained higher than 6 Log CFU/g until the end of ripening, while Enterobacteriaceae became undetectable. Reductions in *L. monocytogenes* counts ranged from 1.6 to 2.2 Log CFU/g in all samples, showing that the Salchichón’s environment was not permissive for its growth. However, the reduction in *L. monocytogenes* was significantly higher in the samples inoculated with *L. sakei* 205. The characterization of the LAB isolates by pulsed field gel electrophoresis (PFGE) showed that *L. sakei* 205 was implanted throughout the ripening time, while in the control group *L. sakei* was not detected, though similar LAB numbers were reached.

*L. sakei* 205 was selected among 182 LAB isolates from dry-cured meat from 3 industries of which 32 inhibited *L. monocytogenes* in vitro. *L. sakei* was the species most frequently showing anti-listerial activity, and other active species were *E. faecium*, *E. hirae*, *L. plantarum*, *Lacticaseibacillus casei* and *L. garviae*. Diversity in the anti-listerial microbiota was observed among producers [109].

Strains of *L. sakei*, *P. pentosaceus*, *S. xylosus* and *S. carnosus* isolated from traditional sausages were able to predominate during the fermentation of Chinese Sichuan sausages, while in the control *Lactobacillus* spp. and *Weissella* prevailed [95]. Undesirable microorganisms such as *Yersinia* spp., *Enterobacter* spp., *Acinetobacter* spp. and *Psychrobacter* spp. were lower in inoculated sausages, where they soon decreased while remaining in high percentages during ripening in samples with spontaneous fermentation. The levels of histamine, putrescine, tyramine and cadaverine were also significantly lower in inoculated samples, being reduced by 83.09%, 69.38%, 51.87% and 57.20%, respectively. The same starter was previously found to lower histamine accumulation by 84.17% in Cantonese sausages. The total volatile basic nitrogen (TVB-N), generally considered as an indicator of freshness and protein degradation in meat, though increased in both inoculated and control samples, was significantly lower in inoculated samples [110].

Rodriguez et al. [96] showed that use of two autochthonous starter cultures, *L. sakei* LS131/*S. equorum* SA25 or *L. sakei* LS131/*S. saprophyticus* SB12, significantly decreased Enterobacteriaceae counts. Both starter cultures significantly increased the α-amino acidic nitrogen (NH2-N), the total basic volatile nitrogen and the free amino acid content, reducing the total BA content by approximately 20%. The presence of *S. saprophyticus* increased the free fatty acid content.

The addition of autochthonous starter cultures able to degrade BAs through oxidative deamination catalyzed by amine oxidases can reduce the accumulation of BAs in fermented sausages. This was observed in a traditional Chinese smoked horsemeat sausage by using strains previously isolated from the product *L. plantarum*, *Ligilactobacillus salivarius* and a combination of the two. These strains also promoted a reduction in the numbers of *Pseudomonas* spp. and Enterobacteriaceae to similar extents. Polymerase chain reaction denaturing gradient gel electrophoresis (PCR DGGE) showed that species present in starter cultures were dominant throughout fermentation, while indigenous microorganisms *Staphylococcus* spp., *S. xylosus*, *S. epidermidis*, *Enterobacter cloacae*, *L. sakei*, *Enterococcus faecium*, *P. pentosaceus*, *Pseudomonas* spp. and *Weissella* spp. showed faint bands. *Pseudomonas* spp. and *Weissella* spp. had disappeared in all batches at day 7. In the control batch indigenous *L. sakei*, *E. faecium* and *P. pentosaceus* remained dominant throughout fermentation. Though the total BA concentration in inoculated batches was significantly lower than in the control, *L. plantarum* and *L. salivarius* did not completely abolish BA accumulation. However, the final levels of total BAs were low, and the lowest histamine concentration at the end of ripening was found in samples inoculated with *L. salivarius* [97].

For autochthonous strains of the yeast species *D. hansenii*, the capacity to increase safety and sensory characteristics has been described. When inoculated on the surface of Salsiccia Sarda, a selected autochthonous strain exerted an anti-mould effect without affecting sensory quality [92]. Peromingo et al. [94] found that two native *D. hansenii* strains reduced the relative expression levels of the *afl*R and *afl*S genes involved in the aflatoxin biosynthetic pathway by *Aspergillus parasiticus* in vitro and reduced the formation of these compounds on fermented sausages. Cebrián et al. [111] reported that an autochthonous strain of *D. hansenii*, alone or in combination with *S. xylosus*, prevented OTA formation by *P. nordicum*, the main OTA producer, in the dry-cured sausage Sauchisson, being therefore proposed as bioprotective culture.

### 4.2. Effects of Autochthonous Cultures on Sensory Attributes of Dry Fermented Sausages

Sensory characteristics were defined by the analysis of products from proteolysis and lipolysis, VOC formation, instrumental analysis of texture and color parameters and panel test assessments. In sensory evaluation, color and texture profile analysis (TPA) parameters of hardness, cohesiveness, springiness, gumminess, elasticity and chewiness were considered.

Franciosa et al. [32] used five selected autochthonous strains from Salame Piemonte, namely two *P. pentosaceus*, two *L. sakei* and one *S. xylosus* strains, already evaluated for safety, in seven combinations in pilot-scale Salame Piemonte production. In sensory evaluation, sausages produced with the *P. pentosaceus* strains were the most appreciated for being less bitter, less acidic and with a more uniform aspect and color intensity. However, all inoculated samples received high odor and color intensity scores and obtained higher scores than the control for the question “Would you buy it?”. Samples produced with *P. pentosaceus* and *L. sakei* were better balanced for most of the investigated attributes. In samples inoculated with *P. pentosaceus* the highest concentrations of acetic acid, diacetyl and acetoin, ethanol, isopentyl alcohol, 1-hexanol and 1-octanol were detected.

Baka et al. [83] found that most evaluated cultures prevented lipid oxidation, as indicated by values of malonaldehyde lower than 1 mg/kg, and that sausages produced with cultures *L. sakei* 4413 and *L. sakei* 8416 had the highest scores for all sensory attributes, namely overall flavor and color, section, external appearance and firmness.

Dias et al. [85] observed that among tested starters, significant differences were only detected for color in sausages inoculated with *S. xylosus* CECT7057 and *L. sakei* CECT7056, that showed more reddish tones. However, control sausages showed the lowest levels of off-flavors and were the most appreciated in overall perception by the panel, although not significantly compared to the inoculated products. Regarding sensory analysis, the association *S. equorum* S2M7/*L. sakei* CV3C2 seemed to have a generally more positive effect than the association *S. xylosus* CECT7057/*L. sakei* CECT7056. Moreover, the inoculated yeast did not show a significant effect in the sensory appreciation, differently from what had been reported by Flores et al. [107] and Corral et al. [52,87].

Hu et al. [89] found that inoculation with *P. pentosaceus* (Pp), *L. curvatus* (Lc), *L. sakei* (Ls) and *S. xylosus* (Sx) autochthonous strains increased the hardness and springiness of Harbin sausage, and the percentages of some aldehydes, ketones, alcohols, acids and esters derived from carbohydrate catabolism, such as 3-hydroxy-2-butanone, from amino acid metabolism, such as 3-phenylpropanol, 2,3-butanediol, phenylethyl alcohol, 2-methyl-propanal, 2-methyl-butanal and 3-methyl-butanal, from β-lipid oxidation, such as 2-pentanone, 2-heptanone and 2-nonanone, and from esterification, such as ethyl esters, were significantly higher in the sausages inoculated with Pp + Sx + Lc than that in the control. Hu et al. [90] reported that *L. plantarum* MDJ2, *L. sakei* HRB10, *L. curvatus* SYS29, *W. hellenica* HRB6 and *L. lactis* HRB0 isolated from Harbin sausage conferred lower a_w_ and pH values on the product and increased LAB counts. Total free amino acids decreased in inoculated samples since they had possibly been converted in VOCs. Electronic nose (E-nose) and sensory analysis indicated that *W. hellenica* HRB6, *L. sakei* HRB10 and *L. curvatus* SYS29 enhanced the pleasant odors. A strong correlation of *W. hellenica* HRB6 and *L. sakei* HRB10 with most of the key VOCs was determined by headspace solid-phase microextraction gas chromatography-mass spectrometry (HS-SPME-GC-MS).

Rodriguez et al. [96] showed that use of two autochthonous starter cultures, *L. sakei* LS131/*S. equorum* SA25 or *L. sakei* LS131/*S. saprophyticus* SB12, slightly but significantly reduced the pH values during the fermentation and increased the formation to nitrosyl-heme pigments with the improvement of the red coloration (a* parameter) and the yellow coloration (b* parameter) determined by colorimetric measurements.

Chen et al. [98] evaluated the role of various single starter strains or strain associations on lipolysis and lipid oxidation in Harbin dry sausages and reported that mixed cultures of *P. pentosaceus*, *L. curvatus*, *L. sakei* and *S. xylosus* modulated lipid degradation, promoting lipid hydrolysis but inhibiting the formation of VOCs derived from lipid autoxidation, thus improving flavor. In another study on Harbin sausage it was found that autochthonous *P. pentosaceus* and *L. curvatus* strains degraded sarcoplasmic proteins, forming higher free amino acid amounts than in the control. *P. pentosaceus* formed the highest amount of VOCs, including aldehydes, alcohols and acids, followed by *L. curvatus* [99].

The autochthonous starter used by Chen et al. [100] increased the amount of the VOC 3-methyl-1-butanol compared to the not-inoculated control and the product inoculated only with *L. plantarum* CMRC6. The starter LS conferred better sensory properties to sausages, showing that the combination of lactobacilli with CNS in the added cultures enhanced sausage quality [100].

The effects of autochthonous LAB strains *L. sakei* BL6, *P. acidilactici* BP2 and *L. fermentum* BL11 in beef jerky were lowering pH, thiobarbituric acid reactive substances (TBARS), indicators of lipid oxidation and carbonyl indicators of protein oxidation. *L. sakei* BL6 produced higher carbohydrate fermentation-derived volatile compound contents, while *P. acidilactici* BP2 produced higher contents of VOCs derived from lipid β-oxidation and amino acid metabolism and showed higher esterase activity. The jerky inoculated with *P. acidilactici* BP2 had the highest acceptability score [20]. The same autochthonous LAB strains were evaluated in the ability to release VOCs by using electronic nose (E-nose) and gas chromatography–ion mobility spectrometry (GC–IMS). All three LAB strains decreased the levels of aldehydes derived from lipid autoxidation, hexanal, heptanal, octanal and nonanal. The strains showed unique flavor profiles. In addition, inoculation with *P. acidilactici* BP2 increased the levels of esters ethyl 3-methylbutanoate, 3-methylbutyl acetate, butyl acetate, ethyl propionate and ethyl acetate. A strong correlation was found between the E-nose and GC–IMS results, so that these methods were both shown to be useful for the selection of autochthonous starter cultures for beef jerky fermentation [102].

Kaban et al. [112] assessed VOC formation for autochthonous strains *L. sakei* S15, *L. plantarum* S24, *L*. *plantarum* S91, *P. pentosaceus* S128b and *S. carnosus* G109 in the Turkish dry fermented sausage Sucuk and observed that most compounds were positively correlated with the products containing only *L. sakei* S15.

Chinese Sichuan sausages inoculated with strains of *L. sakei*, *P. pentosaceus*, *S. xylosus* and *S. carnosus* from traditional products showed less hardness and chewiness, increased springiness and improved color, resulting in better evaluation by a sensory panel. Moreover, in inoculated sausages the concentration of residual nitrite was significantly lower, with a rapid decrease from 150.21 to 4.56 mg/kg vs. 28.81 mg/kg in the control [110].

### 4.3. Autochthonous Probiotics in Dry Fermented Sausages

Since dry fermented sausages are part of the larger group of fermented foods whose regular consumption improves health due to the presence of living, potentially probiotic microorganisms [113,114], they have been analyzed for the occurrence of potential probiotic bacteria as well as evaluations of the suitability of the latter for use as starters in traditional sausages. Klingberg et al. [59] selected three potential probiotic strains, *L. plantarum* MF1291 and MF 1298 and *L. pentosus* MF1300, suitable as starter cultures for the Scandinavian-type fermented sausages that were identified among 22 dominant non-starter lactic acid bacteria (NSLAB) well-adapted to Norwegian and Swedish fermented meats. These were able to lower pH below 5.1 in a meat model, survive at pH 2.5 and in presence of 0.3% oxgall, adhere to the human colon adenocarcinoma cell line Caco-2 and express antimicrobial activity against potential pathogens such as *Escherichia coli*, *B. cereus*, *Shigella flexneri*, *Y. enterocolitica*, *Salmonella* Typhimurium and *L. monocytogenes*. When applied as starter cultures for the production of a Norwegian salami according to the traditional recipe, these cultures reached high viable counts and pH values between 4.8 and 4.9 at the end of ripening. The sensory characteristics of the sausage inoculated with the autochthonous starter were comparable to those of sausages manufactured with the commercial meat starter culture *L. curvatus* HJ5, according to the evaluation of 19 parameters.

Pavli et al. [101] evaluated the performance of *L. plantarum* L125, a strain with probiotic potential isolated from a traditional Greek fermented sausage, as an adjunct culture for the production of dry-fermented pork sausages. The strain was inoculated together with a commercial starter culture and remained at high levels until the end of ripening without affecting sensory parameters.

More recently, Dincer et al. [115] reported the probiotic potential of *L. sakei* isolated from dry fermented Pastırma beef sausage. Two strains, *L. sakei* 8.P1 and 8.P2, produced proteinaceous compounds able to inhibit *L. monocytogenes*, *S. aureus* and *P. aeruginosa* and tolerated simulated gastric juice conditions, showing a good cell adhesion capacity.

## 5. Conclusions

This analysis of studies regarding the improvement of traditional fermented meat products by using autochthonous cultures highlighted multiple advantages obtained both in safety and in sensory quality. Therefore, studies on the biodiversity of the autochthonous microorganisms should be followed by the definition of safety and health promoting attitudes and technologically relevant features of the microbial isolates to devise appropriate inoculation schemes with native cultures that can ensure safety and stabilize quality. There is still much to disclose regarding the numerous traditional production processes not characterized microbiologically, and efforts should be devoted to selecting native bacteria suitable for use as starter cultures. Moreover, proving that a moderate consumption of these products enriched with native probiotic microorganisms can benefit health could incentivize the continuation of manufacturing traditions in marginal areas. Indeed, the probiotic potential of native LAB from traditional fermented sausages is still little explored and could lead to an increased benefit on health from the consumption of these products.

Many of the studies summarized in this review showed that the natural microbiota include species with the potential to prevent the development of hazardous microorganisms and substances, so their selection and application can represent an efficient tool to increase consumers’ trust and appreciation for traditional fermented meat products guaranteeing constant quality, safety, sensory distinctness and delivery of beneficial microorganisms.

## Figures and Tables

**Figure 1 microorganisms-11-01306-f001:**
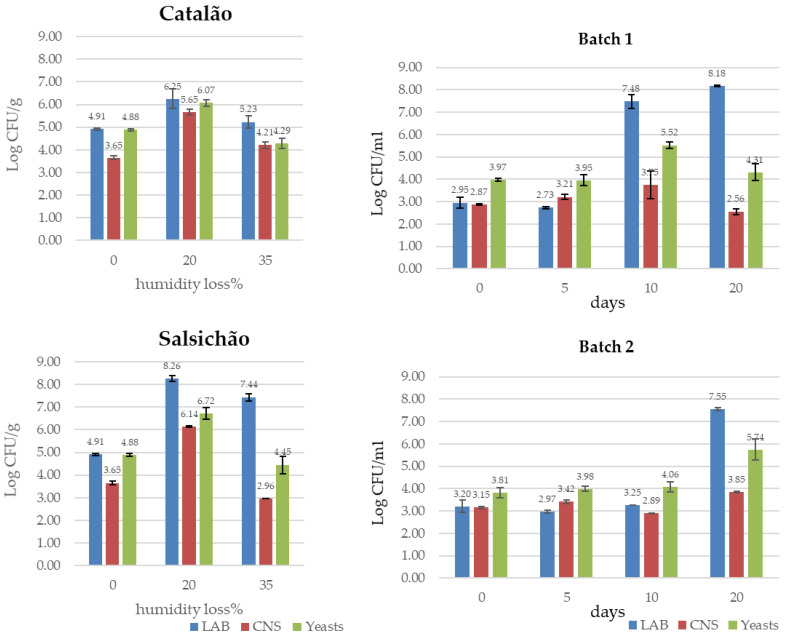
Natural evolution of LAB, CNS and yeasts in Catalão, Salsichão and Ciauscolo sausages as reported by Laranjo et al. [27] and Belleggia et al. [56]. A weight loss of 35%, corresponding to humidity loss, was reached in about 18 days for “Catalão” and in 35 days for “Salsichão”.

**Table 1 microorganisms-11-01306-t001:** Effect of autochthonous cultures on the safety of traditional fermented meat products.

Microbial Strains	Product	Effect	Reference *
*S. simulans* QB7	Qianwufu fermented sausage, Guizhou province, China	reduced growth of undesirable bacteria	Li et al. [44]
*L. curvatus* 8427, *L. plantarum* 7423, *L. sakei* 8416, 4413 and 8426	Greek fermented sausages	inhibition of undesirable microorganisms	Baka et al. [83]
*L. curvatus* 54M16	fermented sausages of Campania region, Italy	lower numbers of *Enterobacteriaceae*	Casaburi et al. [84]
*L. sakei* CV3C2 and CECT7056, *S. equorum* S2M7, *S. xylosus* CECT7057, yeast strain 2RB4	Painho da Beira Baixa, Portugal	decrease in pH, *Enterobacteriaceae*, *L. monocytogenes* and total Bas	Dias et al. [85]
*L. sakei* CV3C2, *S. equorum* S2M7, yeast 2RB4	Paio do Alentejo, Portugal	decrease in *L. monocytogenes* counts and vasoactive amines tryptamine and β-phenylethylamine content	Dias et al. [86]
*L. sakei* CV3C2 and CECT7056, *S. equorum* S2M7, *S. xylosus* CECT7057, yeast strain 2RB4	Paio do Alentejo, Portugal	decrease in pH, *L. monocytogenes* counts and total BA content	Dias et al. [87]
*L. sakei*, *S. epidermidis*	Harbin sausage, China	decrease in tyramine content	Dong et al. [78]
*L. curvatus* 54M16	fermented sausages of Campania region, Italy	total inhibition of *L. monocytogenes* native from raw materials, inhibition of *Brochothrix*, *Psychrobacter*, *Pseudomonas* and *Enterobacteriaceae*	Giello et al. [88]
*L. curvatus*, *L. sakei*, *P. pentosaceus*, *S. xylosus*	Harbin dry sausage, China	decrease in a_w_	Hu et al. [89]
*L. curvatus* SYS29, *L. lactis* HRB0, *L. plantarum* MDJ2, *L. sakei* HRB10, *W. hellenica* HRB6	traditional dry sausage, China	decrease in pH and a_w_, increase in LAB counts	Hu et al. [90]
*L. plantarum* S50, S51, S72, S74, S85	Sucuk, Turkey	Inhibition of *L. monocytogenes*, rapid decrease in pH	Kamiloglu et al. [91]
*D. hansenii*	Salsiccia Sarda, Italy	Anti-mold effect	Murgia et al. [92]
*L. sakei* 205	Salchichón, Spain	decrease in *L. monocytogenes* counts	Martín et al. [93]
*D. hansenii*	dry-cured meat products	decrease in aflatoxin formation by *Aspergillus parasiticus*	Peromingo et al. [94]
*L. sakei*, *P. pentosaceus*, *S. carnosus*, *S. xylosus*	Sichuan sausage, China	decreased levels of histamine, putrescine, tyramine, cadaverine and residual nitrites	Ren et al. [95]
*L. sakei/S. equorum* SA25 *L. sakei* LS131/*S. saprophyticus* SB12	Galician Chorizo, Spain	pH decrease, increase in free amino acids and decrease of total BAs by approximately 20%	Rodríguez et al. [96]
*L. plantarum*, *L. salivarius*	traditional smoked horsemeat sausage, China	decrease in all indigenous microorganisms, including *Enterobacter cloacae*, *Enterococcus faecium*, *Pseudomonas* spp. and *Weissella* and in total BAs and histamine	Zhang et al. [97]

* literature sources are alphabetically ordered unless cited in previous sections.

**Table 2 microorganisms-11-01306-t002:** Effect of autochthonous cultures on the sensory quality and composition of traditional fermented meat products.

Microbial Strains	Product	Effect	Reference *
*L. fermentum* BL11, *L. sakei* BL6, *P. acidilactici* BP2	Beef jerky, China	Lower pH and indicators of lipid and protein oxidation, higher VOC formation from carbohydrates for *L. sakei*; higher VOC formation from lipid β-oxidation and amino acid metabolism, esterase activity and acceptability score for *P. acidilactici*	Wen et al. [20]
*S. simulans* QB7	Qianwufu fermented sausage, Guizhou province, China	reduced growth of undesirable bacteria	Li et al. [44]
*D. hansenii* M4 and P2	dry-cured fermented sausages	strain P2 decreased lipid oxidation and increased acid compounds, strain M4 increased sulphur containing compounds, no differences in consumer acceptance	Cano-García et al. [50]
*L. sakei*, *P. pentosaceus*, *S. xylosus*	Salame Piemonte, Italy	improvement of the sensory properties	Franciosa [32]
*L. plantarum* MF1291 and MF 1298, *L. pentosus* MF1300	Traditional Norwegian salami	19 sensory parameters comparable to the commercial starter culture *L. sakei* HJ5	Klingberg et al. [59]
*L. curvatus* 8427, *L. plantarum* 7423, *L. sakei* 8416, 4413 and 8426	Greek fermented sausages	prevention of lipid oxidation; higher scores for all sensory attributes	Baka et al. [83]
*L. curvatus 54M16*	fermented sausages of Campania region, Italy	more intense ripened flavor	Casaburi et al. [84]
*L. sakei* CV3C2 and CECT7056, *S. equorum* S2M7, *S. xylosus* CECT7057, yeast strain 2RB4	Painho da Beira Baixa, Portugal	higher scores in sensory attributes	Dias et al. [85]
*L. sakei* CV3C2 and CECT7056, *S. equorum* S2M7, *S. xylosus* CECT7057, yeast strain 2RB4	Paio do Alentejo, Portugal	negative effect on the sensory characteristics of fermented sausages	Dias et al. [86]
*L. curvatus*, *L. sakei*, *P. pentosaceus*, *S. xylosus*	Harbin dry sausage, China	increase in hardness and springiness, higher percentages of aldehydes, ketones, alcohols, acids and esters	Hu et al. [89]
*L. curvatus* SYS29, *L. lactis* HRB0, *L. plantarum* MDJ2, *L. sakei* HRB10, *W. hellenica* HRB6	traditional dry sausage, China	increase in VOC content, decrease in total content of free amino acids, enrichment of pleasant odors for *L. sakei* and *W. hellenica*	Hu et al. [90]
*L. sakei*, *P. pentosaceus*, *S. carnosus*, *S. xylosus*	Sichuan sausage, China	lower hardness and chewiness, increased springiness; improved color and sensory attributes	Ren et al. [95]
*L. sakei* LS131/*S. equorum* SA25 or *L. sakei* LS131/*S. saprophyticus* SB12	Galician Chorizo, Spain	increment in the α-amino acid nitrogen, total basic volatile nitrogen and free amino acids, improvement of color	Rodríguez et al. [96]
*L. brevis* R4, *L. curvatus* R5, *L. fermentum* R6, *P. pentosaceus* R1	Harbin dry sausage	*P. pentosaceus* hydroxyl radical and 1,1-diphenyl-2-picrylhydrazyl (DPPH) radical scavenging activity, inhibition of lipid peroxidation, high SOD and glutathione peroxidase (GSH-Px) activities	Chen et al. [98]
*L. brevis* R4, *L. curvatus* R5, *L. fermentum* R6, *P. pentosaceus* R1	Harbin dry sausage	*P. pentosaceus* strongest proteolysis activity, highest formation of soluble peptides and free amino acids, VOCs from sarcoplasmic proteins	Chen et al. [99]
*S. xylosus* SX16, *L. plantarum* CMRC6	Chinese Dong fermented pork (Nanx Wudl)	acceleration of acidification and proteolysis, increase in total free amino acids and essential free amino acids Phe, Ile and Leu, increase in 3-methyl-1-butanol	Chen et al. [100]
*L. plantarum* L125	traditional Greek dry-fermented sausage	desiderable technological characteristics	Pavli et al. [101]
*L. fermentum* BL11, *L. sakei* BL6, *P. acidilactici* BP2	Beef jerky, China	decrease in lipid autoxidation-derived aldehydes; increment of esters for *P. acidilactici*; strain-specific flavour profiles	Wen et al. [102]

* literature sources are alphabetically ordered unless cited in previous sections.

## Data Availability

Data sharing not applicable.

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
