# Peer review of "Autochthonous Cultures to Improve Safety and Standardize Quality of Traditional Dry Fermented Meats"

_microorganisms, 2023, doi:10.3390/microorganisms11051306_

Round 1

Reviewer 1 Report

Manuscript ID: microorganisms-2294463 Autochthonous cultures to improve safety and stabilize quality of traditional dry fermented meats

General opinion

Thank you for entrusting me with a review of the manuscript. The authors conducted an analysis of the available scientific research results and legal regulations regarding traditional dry - fermented meat products. They paid special attention to the importance of autochthonous cultures in the production of these meat products. The manuscript presented for assessment is a valuable source of knowledge not only about the desired microorganisms - starter cultures, but also about a competitive, unwanted microflora. The authors tried to show a wide spectrum of factors affecting the safety of traditional dry meat products, among others Issues of use of red meat for production, using nitrates and forming biogenic amines. The manuscript is a reliable review of the topic, mostly based on the results of research published in the last years. However, there are very extensive fragments containing detailed information, and very extensive fragments containing very general information. This means that the manuscript is generally very extensive. If the authors could compress the manuscript's content, I think it would be beneficial. I think that after small corrections, the manuscript should be accepted.

Detailed Comments:

Line 44 – „..oligoelements…” - explain what you meant

Line 328 – „…nitrates/nitrites from animal sources…” - animals are not a source of nitrates, correct this sentence

Author Response

C1. General opinion

Thank you for entrusting me with a review of the manuscript. The authors conducted an analysis of the available scientific research results and legal regulations regarding traditional dry - fermented meat products. They paid special attention to the importance of autochthonous cultures in the production of these meat products. The manuscript presented for assessment is a valuable source of knowledge not only about the desired microorganisms - starter cultures, but also about a competitive, unwanted microflora. The authors tried to show a wide spectrum of factors affecting the safety of traditional dry meat products, among others Issues of use of red meat for production, using nitrates and forming biogenic amines. The manuscript is a reliable review of the topic, mostly based on the results of research published in the last years. However, there are very extensive fragments containing detailed information, and very extensive fragments containing very general information. This means that the manuscript is generally very extensive. If the authors could compress the manuscript's content, I think it would be beneficial. I think that after small corrections, the manuscript should be accepted.

R1. We are grateful to the reviewer for appreciating our manuscript. The manuscript was read again carefully and the parts not conveying useful information were shortened. Some details, such as numeric values, were deleted unless necessary to illustrate autochthonous culture effects.

C2. Detailed Comments:Line 44 – „..oligoelements…” - explain what you meant

R2. The term was changed in micro-nutrients (Line 40), i.e. nutrients essential in trace amounts

C3. Line 328 – „…nitrates/nitrites from animal sources…” - animals are not a source of nitrates, correct this sentence

R3. The expression was changed with “meat” (Line 298)

Reviewer 2 Report

Dear, considering the manuscript entitled Autochthonous cultures to improve safety and stabilize quality of traditional dry fermented meats, submitted for review, my opinion is as follows.

The manuscript is very confusing and not well structurally organized, thus requiring a lot of effort to read and understand what the authors actually want to do/achieve. What are the main objectives of the manuscript, which will be further discussed? 

In its current form, the manuscript is not readable and does not possess interest for readers. 

The aim and scope of the review do not clear, followed by further sections. 

Overall, a manuscript must be improved significantly.

Author Response

C1. Dear, considering the manuscript entitled Autochthonous cultures to improve safety and stabilize quality of traditional dry fermented meats, submitted for review, my opinion is as follows.

The manuscript is very confusing and not well structurally organized, thus requiring a lot of effort to read and understand what the authors actually want to do/achieve. What are the main objectives of the manuscript, which will be further discussed? 

In its current form, the manuscript is not readable and does not possess interest for readers. 

R1. The manuscript was re-checked for clarity. The succession of sections remained unvaried since it was chosen to illustrate first the characteristics of traditional dry fermented meat products, their significance for the producing countries, protection policies and safety concerns in order to allow the evaluation of how use of autochthonous cultures can introduce improvements in their characteristics and preservation of the traditional production processes.

C2. The aim and scope of the review do not clear, followed by further sections. 

Overall, a manuscript must be improved significantly.

R2. The scope was remarked at lines 18-21 and 364-368.

Round 2

Reviewer 2 Report

Dear, following the revised version submitted for a second revision round, I think that despite the authors' efforts to improve the manuscript, it is not readable yet.

The title of the manuscript can be improved, instead, word stabilize use standardized.

Despite well definied section, the main shortcoming and flaws are related to the structure of the manuscript.

The whole length of the manuscript is 23 pages. Under such circumstances, the manuscript must be well organized, and require excellent skills from the authors to keep the attention/concentration of readers. In its current form, manuscript require effort to read and understand, and thus must be improved.

The section Introduction is unnecessarily long, and the study’s aim and scope are still missing. Section Introduction can be improved and organized in a logical order which briefly and logically leads the reader to your research, aim,s and the scope of the study.

Section Technologically relevant microorganisms in dry fermented meats must be improved. Could you divide the section into subsections related to the technological properties of relevant microorganisms ie. their sensory, quality, and safety impact?

Also, the section Diversity of naturally occurring technologically relevant microorganisms in dry fermented meats can be improved in a way to divide into the subsection, depending on the meat products or other relevant factors.

Safety concerns in traditional fermented dry meats can be divided into subsections, addressing main microbiological hazards (toxins) and/or chemical hazards.

The effect of autochthonous microbial cultures on the safety and quality of dry fermented sausages can be divided into subsections to better reflect their impact on the mentioned characteristics.

The whole manuscript is not yet ready for publishing and thus must be improved. The main shortcoming and flaws are related to the structure and organization of the sections of the manuscript. Each of the sections must be divided into subsections depending on the covered topics. 

Author Response

Response to reviewer 2

In this response C stand for the original comment and A for our answer

C1. Dear, following the revised version submitted for a second revision round, I think that despite the authors' efforts to improve the manuscript, it is not readable yet.

The title of the manuscript can be improved, instead, word stabilize use standardized.

A1. The title was modified as suggested

C2. Despite well definied section, the main shortcoming and flaws are related to the structure of the manuscript.

The whole length of the manuscript is 23 pages. Under such circumstances, the manuscript must be well organized, and require excellent skills from the authors to keep the attention/concentration of readers. In its current form, manuscript require effort to read and understand, and thus must be improved.

The section Introduction is unnecessarily long, and the study’s aim and scope are still missing. Section Introduction can be improved and organized in a logical order which briefly and logically leads the reader to your research, aim,s and the scope of the study.

A2. The Introduction was somewhat shortened, but, since it presents type of products and protection policies all information was retained. Aims and scopes were transferred at the end of this section (Lines 135-141). Now the whole manuscript is 21 pages.

C3. Section Technologically relevant microorganisms in dry fermented meats must be improved. Could you divide the section into subsections related to the technological properties of relevant microorganisms ie. their sensory, quality, and safety impact?

A3. Four sub-sections were created and section 3 was combined with section 2 (Lines 147, 177, 215, 225).

C4. Also, the section Diversity of naturally occurring technologically relevant microorganisms in dry fermented meats can be improved in a way to divide into the subsection, depending on the meat products or other relevant factors.

A4. This section was already very short and therefore it was not split. Instead it was combined with section 2 to complete the list of microorganisms occurring in dry fermented sausages (section 2.4).

C5. Safety concerns in traditional fermented dry meats can be divided into subsections, addressing main microbiological hazards (toxins) and/or chemical hazards.

A5. Subsections were created according to the hazard type (Lines 261, 288, 327, 348, 352).

C6. The effect of autochthonous microbial cultures on the safety and quality of dry fermented sausages can be divided into subsections to better reflect their impact on the mentioned characteristics.

A6. The division in 3 subsections was done: effects of autochthonous cultures on i safety (Line 389), on sensory improvement (Line 539), and probiotic characteristics (Line 628).

C7. The whole manuscript is not yet ready for publishing and thus must be improved. The main shortcoming and flaws are related to the structure and organization of the sections of the manuscript. Each of the sections must be divided into subsections depending on the covered topics. 

A7. We hope to have succeeded in improving the manuscript according to these indications.